# Preserving Knowledge Invariance: Rethinking Robustness Evaluation of Open Information Extraction

**Ji Qi[1], Chuchun Zhang[2], Xiaozhi Wang[1], Kaisheng Zeng[1],**
**Jifan Yu[1], Jinxin Liu[1], Lei Hou[1], Juanzi Li[1], Bin Xu[1*]**
[1]Department of Computer Science and Technology, BNRist, Tsinghua University
[2]University of International Business and Economics
qj20@mails.tsinghua.edu.cn, 202025003@uibe.edu.cn

## Abstract

The robustness to distribution changes ensures that NLP models can be successfully applied in the realistic world, especially for information extraction tasks. However, most prior evaluation benchmarks have been devoted to validating pairwise matching correctness, ignoring the crucial validation of robustness. In this paper, we present the first benchmark that simulates the evaluation of open information extraction models in the real world, where the syntactic and expressive distributions under the same knowledge meaning may drift variously. We design and annotate a large-scale testbed in which each example is a knowledge-invariant clique that consists of sentences with structured knowledge of the same meaning but with different syntactic and expressive forms. By further elaborating the robustness metric, a model is judged to be robust if its performance is consistently accurate on the overall cliques. We perform experiments on typical models published in the last decade as well as a representative large language model, and the results show that the existing successful models exhibit a frustrating degradation, with a maximum drop of 23.43 $F_1$ score. Our resources and code are available at https://github.com/qijimrc/ROBUST.

## 1 Introduction

Open Information Extraction (OpenIE) aims to extract $n$-ary knowledge tuples $\{(a_1, p, a_2, ..., a_n)\}$ consisting of $n$ arguments and one predicate from the natural text in a domain-independent manner, which has been served as the backbone to benefit NLP applications for many years (Liu et al., 2021; Pei et al., 2022; Chen et al., 2021).

Due to its structural flexibility, the evaluation of OpenIE is a nontrivial problem, which in turn drives the advancement of the task. Early studies (Stanovsky and Dagan, 2016; Zhan and Zhao, 2020) measure the performance of extractions

---

*Corresponding author: xubin@tsinghua.edu.cn

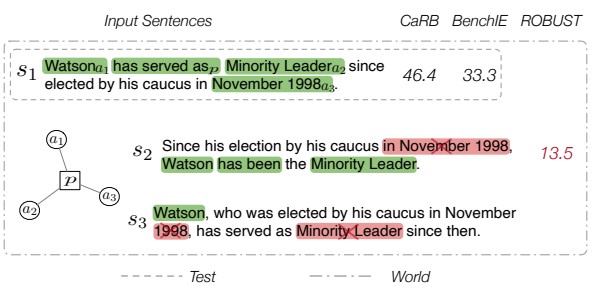

Figure 1: Extraction results of OpenIE6 for three semantically equivalent sentences from CaRB and ROBUST. The proposed benchmark ROBUST computes the robustness score on a clique of sentences.

based on the lexical matching of syntactic heads between elements. To tackle the overly lenient metric, subsequent approaches (Lechelle et al., 2019; Bhardwaj et al., 2019; Gashteovski et al., 2022) propose to use of exact matching between tokens for delicate evaluation. Among these benchmarks, CaRB (Bhardwaj et al., 2019) adopts the all-pair matching table to compute the tuple match scores between extractions, which has been considered the de facto standard for evaluation. Research including these efforts has been devoted to evaluating the pairwise matching correctness between model extractions and golden facts on a sentence.

However, the conventional evaluation benchmarks do not measure the robustness of models in the realistic open-world scenario, where the syntactic and expressive forms may vary under the same knowledge meaning (Qi et al., 2023). As shown in Figure 1, while the three sentences $s_1, s_2, s_3$ contain the same structured knowledge $(a_1, p, a_2, a_3)$, the state-of-the-art model OpenIE6 successfully extracts facts (in green color) on sentence $s_1$, but fails to predict arguments (in red color) on the other sentences due to the syntactic and expressive drifts. In this example, the sentence $s_1$ comes from CaRB which has a similar syntactic distribution to the training set, and existing benchmarks can only eval-

uate models on this limited target attributing it the commendable scores (46.4/33.3), rather than on the other world samples. For accurate and faithful evaluation, we should measure the performance of models on sentences with various syntactic and expressive distributions under the same knowledge meaning (Zhong et al., 2022).

Nevertheless, it is not trivial to construct a benchmark that satisfies the aforementioned conditions of encompassing both knowledge invariance and distributional shift. First, manual annotation of parallel texts to maintain the same knowledge meaning with different syntactic and expressive forms may result in either too trivial or artificial. Second, it is difficult to build a metric that measures the robustness as well as be compatible with existing benchmarks (e.g., (Bhardwaj et al., 2019; Gashteovski et al., 2022)) to ensure comparability.

On the other hand, natural language paraphrasing is defined as producing sentences with different surface forms (syntactic and lexical) by conveying the same semantic meaning (Zhou and Bhat, 2021). Going beyond the pairwise correctness comparison, can we evaluate the robustness of models based on reliable paraphrases equipped with syntactic and expressive transformations?

In this paper, we introduce ROBUST, a **R**obust **O**penIE **B**enchmark with **U**biquitous **S**yntactic **T**ransformations, aiming to evaluate the robustness of OpenIE models. ROBUST is a large-scale human-annotated benchmark consisting of $1,272$ robustness testing cliques, where each clique contains sentences with different syntactic and expressive variations while conveying the same underlying knowledge meaning, for a total of 4,971 sentences and 16,191 knowledge extractions. To obtain each clique, we first adopt a syntactically controllable paraphraser with diversified syntactic sampling and expressive filtering strategies to generate paraphrases for each sentence in CaRB. We then design a two-stage annotation pipeline to perform sentence correction and knowledge extraction for each individual paraphrase in cliques based on human experts. This data paradigm enables evaluation to go beyond pairwise matching to clique-wise comparisons. Upon the testbed structure, we calculate the robustness scores with respect to the worst performance within a clique and further analyze the performance variances on all cliques. This metric fairly reflects the robustness of models to distributional drifts and is also compatible with existing benchmarks calculated at one sentence magnitude.

To explore the robustness of existing models, we implement typical OpenIE systems published in the past decade. The experimental results show a dramatic degradation in model performance on ROBUST, with an average drop of 18 percentage points in $F_1$ scores, indicating that the robustness of existing successful models is far from satisfactory. We then further analyze the correlation between the variances of the model performance and the divergences of the syntactic distances on the cliques. The results find that the variance grows as the syntactic distance increases, and models behaved with similar variance on most of the cliques also demonstrate the inner consistency of our benchmark. In addition, we also evaluate the a representative large language model ChatGPT[1] for OpenIE. Experimental results demonstrate that ChatGPT achieves a remarkable performance that is compatible with the state-of-the-art model on CaRB ($F_1$ score of 0.516 under the 10-shot setting), yet it still exhibits the robustness issue on ROBUST ($F_1$ score of 0.275 under the 10-shot setting).

## 2 The ROBUST Benchmark

In this section, we describe the details of the benchmark construction. The benchmark consists of cliques based on syntactically diverse paraphrase generation and human annotation to unsure the knowledge invariance and distributional shift, where both the syntactic transformations sampled from real world and the human experience guarantee the naturalness. We also provide details of annotations and strategies in the Appendix A.1 and A.2.

### 2.1 Data Preparation

**Paraphrase Generation.** Considering the compatibility with previous benchmarks, we build our benchmark based on CaRB (Bhardwaj et al., 2019), which contains 1,272 sentences[2] of general domain originated from OIE2016 (Stanovsky and Dagan, 2016) with high-quality n-tuples annotations. To build sufficient paraphrases, we adopt AESOP (Sun et al., 2021), a syntactically controllable paraphrasing model generating paraphrases by specifying pruned target syntactic trees that can be sampled diversely. The model used in our work is trained on a parallel annotated data with two-level target

---

[1]https://chat.openai.com/
[2]We remove 10 sentences that do not have extractions from the original data.

| | Sentence | Syntax | Knowledge | Extractions | CaRB | ROBUST |
|---|---|---|---|---|---|---|
| $s_1$ | Watson has served as Minority Leader since elected by his caucus in November 1998. | | | <Wastson, has served as, Minority Leader, in November 1998> | ✓ | ✓ |
| $s_2$ clique | Since his election by his caucus in November 1998, Watson has been the Minority Leader. | | | <Wastson, has been, Minority Leader, since his election by his caucus in November 1998> | ✗ | ✓ |
| $s_3$ | Watson, who was elected by his caucus in November 1998, has served as Minority Leader since then. | | | <Wastson, was elected by, his caucus in November 1998> <Wastson, has served as, Minority Leader, since then> | ✗ | ✓ |

Figure 2: An example of a robustness clique consisting of three sentences from ROBUST, where the sentences exhibit syntactic and expressive variants while preserving the same structured knowledge meaning. In contrast to conventional metrics, ROBUST measures the robustness score on a clique of all nodes.

syntactic trees. During generation, we first collect a set of constituency parse pairs $\{(T^{\mathcal{P}}_{s_i}, T^{\mathcal{P}}_{t_i})\}$ pruned at height 3 from the ParaNMT-50M (Wieting and Gimpel, 2018). And then for each sentence $s$ with its constituency parse tree $T$, we obtain 2 most similar parses $\{(T'^{\mathcal{P}}_{s_i}, T'^{\mathcal{P}}_{s_2})\}$ by calculating weighted ROUGE scores between parse strings and select 5 top-ranked parses from $\{T^{\mathcal{P}}_{t_i}\}$ for each $T'^{\mathcal{P}}_{s_i}$ by a sampling with the distribution of $T^{\mathcal{P}}_t \sim \frac{(\#T'^{\mathcal{P}}_{s_i}, T^{\mathcal{P}}_t)}{\sum_j \#(T'^{\mathcal{P}}_{s_i}, T'^{\mathcal{P}}_{t_j})}$. We thus generate 10 syntactically varying paraphrases for each sentence.

**Diversified Expressive Filtering.** Though different syntactic trees are specified in the paraphrase generation, we find that there are still similar expressions in the generated sentences. Therefore, we further filter the paraphrases with a heuristic search strategy to maintain the most diverse ones. For each clique composed of multiple sentence nodes, including an original sentence and multiple paraphrases, we first calculate the BLEU scores (Papineni et al., 2002) between all pairs of nodes. We then repeat the following simple strategy on paraphrase nodes until reaching the maximum acceptable number to eliminate homogeneity: (1) find the pair of nodes with the largest score in the current clique; (2) remove a node if its length is less than 2/3 of the original sentence, otherwise remove the node with the highest sum of scores with all other nodes. As depicted in Figure 1, the remaining sentences $s_2$ and $s_3$ exhibit distinct syntactic structures and expressive forms compared to the original sentence $s_1$. The detailed process with an example is shown in Appendix A.2.2.

## 2.2 Annotation

For each paraphrase within a clique, we further design a two-stage annotation pipeline based on human experts to perform sentence correction and structured knowledge extraction. All annotators undergo training with tutorials to pass a final examination, and our batch-wise sampling validation ensure an annotation accuracy of over 90%. Detailed annotation including annotators, platforms, and quality checking can be found in Appendix A.1.

**Paraphrase Annotation.** While automatically generated paraphrases present syntactic and expressive variants, the correctness of the sentences cannot be fully guaranteed. To ensure the quality of the sentences, we perform a thorough paraphrase annotation with three types of corrections:

- **Grammar Correcting**: Correct grammatical mistakes in sentences to ensure the fluency.

- **Phrase Replacing**: Replace the incorrect phrases in sentences to ensure the correctness.

- **Sentence Rewriting**: Rewrite the entire sentence if it has a semantic difference from the original sentence.

All operations are required to preserve both the distinctiveness of the annotation from the original sentence and their semantic equivalence. Based on this paradigm, all paraphrases are guaranteed to differ from the original sentence in expression, while retaining the same semantic meaning. As shown in Figure 2, the three sentences in the 1st column exhibit different syntactic and expressive forms. A detailed process is available in Appendix A.1.1.

| Benchmark | #Sent. | #Extr. | #Cliques | Source |
|-----------|--------|--------|----------|--------|
| OIE2016 | 3,200 | 10,359 | – | WSJ,Wiki |
| Re-OIE2016 | 600 | | – | OIE2016 |
| CaRB | 1,272 | 5,262 | – | OIE2016 |
| BenchIE | 300 | 136,357 | – | CaRB$_{part}$ |
| **ROBUST** | 4,971 | 16,191 | 1,272 | CaRB$_{full}$ |

Table 1: Quantitative statistics of ROBUST. (Sent.: sentence, Extr.: extraction)

**Knowledge Annotation.** In the second stage, we leverage human experts to annotate N-ary knowledge tuples on the paraphrases finished in the first stage. We design a guideline involving an iterative process to instruct annotators in extracting all possible facts from a sentence. By referring to the annotation of CaRB, in each iteration, we also divide the task of annotating into three steps: (1) recognizing the predicate, (2) finding the arguments for that predicate, and (3) optionally obtaining the time and location arguments for the tuple if possible.

In particular, we distribute the complete clique to individual annotators to obtain extractions with the same structured knowledge meaning. This annotation process ensures the characteristics in CaRB (i.e. Completeness, Assertedness, Informativeness, and Atomicity) while maintaining consistency with the underlying knowledge. As illustrated in the fourth column of Figure 2, the extractions from different sentences correspond to the same underlying knowledge. Detailed annotation process is available in Appendix A.1.2.

## 3 Data Analysis

To understand the general characteristics of ROBUST, we provide quantitative statistics at different granularities in comparison to previous benchmarks. In contrast to the traditional analysis on words and sentences, we further investigate the syntactic phenomena on cliques to explain the robustness evaluation.

### 3.1 Data Statistics

Table 1 shows the quantitative statistics of ROBUST and representative OpenIE benchmarks, including OIE2016 (Stanovsky and Dagan, 2016), Re-OIE2016 (Zhan and Zhao, 2020), CaRB (Bhardwaj et al., 2019) and BenchIE (Gashteovski et al., 2022). In comparison with the conventional dataset, ROBUST provides the largest number of human-annotated high-quality sentences. Meanwhile,

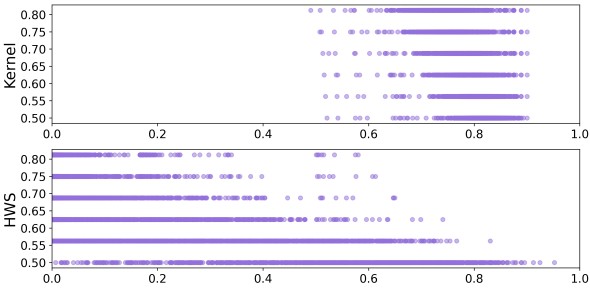

Figure 3: The average syntactic distances/similarity in each clique is calculated using HWS distance and Convolutional Tree Kernels, where the x-axis refers to the hierarchical discounting weights for two algorithms.

based on the annotation paradigm, ROBUST raises a new data structure, the clique, which establishes the interconnection of sentences with underlying knowledge. The average number of sentences per clique is 3.877.

In addition, we find that previous benchmarks completely originate from OIE2016 based on Wiki and newswires, potentially leading to distribution bias to similar training corpus, especially for pre-trained language models (e.g. BERT (Devlin et al., 2019)) trained on the general corpora. ROBUST mitigates this bias by extending syntactic and expressive distributions to realistic scenarios. We further compute the vocabulary sizes for CaRB and ROBUST, resulting in 7648 and 7981, respectively, demonstrating that our natural annotations do not introduce many rare words.

### 3.2 Syntactic Analysis

The proposed benchmark measures the robustness of models on the drifts of linguistic observations. Therefore, the syntactic divergence in the clique is the key to ensuring robustness evaluation. We provide a thorough syntactic analysis of cliques to investigate the divergence.

**Metrics of Syntactic Correlation.** In order to analyze the syntactic divergence in the cliques, we need a metric to measure the syntactic correlation between two sentences. A fast and effective algorithm is the HWS distance proposed in (Qi et al., 2023), which calculates the syntactic tree distance between two sentences based on a hierarchically weighted matching strategy, where smaller weights imply a greater focus on the comparison of skeletons. The value domain of this is $[0, 1]$, where 1 indicates the farthest distance. However, we find that their method may lead to the overcounting

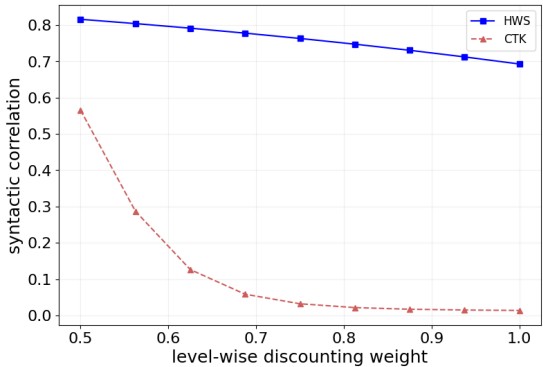

Figure 4: The average syntactic distance/similarity over all cliques with the hierarchical discounting weights. Cliques containing only one point will be a line with a value of 0 or 1.

problem for repeated consecutive spans [3]. We revise the original algorithm to solve the problem while maintaining efficiency. The details of the revised algorithm are shown in Appendix A.2.1 for ease of use.

We additionally implement the algorithm of Convolutional Tree Kernel (CTK) similarity proposed in (Collins and Duffy, 2001) to fairly illustrate the syntactic phenomenon. In contrast to distance, it measures the similarity between a pair of tree structures by counting the number of tree fragments in common. The value domain of this algorithm is also $[0, 1]$, where $1$ means the maximum similarity.

**Intra-clique Syntactically Analysis.** To exhaustively investigate the syntactic divergence on the cliques, we calculate the average syntactic distance/similarity in each individual clique based on the algorithms described above. The result is shown in Figure 3, where the horizontal axis and vertical axis are the output and the discounting weights of two algorithms, respectively.

Overall, we observe that the values of syntactic distance and syntactic similarity are mainly scattered between $[0.6, 0.9]$ and $[0.0, 0.7]$, respectively, indicating that most of the cliques exhibit significant syntactic discrepancies. Another notable observation is that the distribution of the HWS scatter representing the distance is closer to 1 as the discount weight decreases, suggesting that the differences in syntactic skeletons are more significant in ROBUST.

**Inter-cliques Syntactically Analysis.** Going be-

---



[3] For two strings $s_1 s_3 s_4$ and $s_1 s_2 s_1$ with consecutive span $s_1$ in common (e.g, SVPNP and SVPNPVP), the resulting distance may increase with the repetition of span $s_1$.

yond the individual clique, we further explore the syntactic divergence over all cliques. As shown in Figure 4, we average the mean of clique-wise syntactic distance/similarity on all cliques, based on the linearly increased discounting weights. We find that the average similarity of syntactic trees on ROBUST decreases rapidly as the discounted weight of the algorithm increases. Considering that increasing the weights implies a reduced focus on the low-level tree fragments, this result suggests that ROBUST involves prominent variability in the high-level skeleton of syntactic trees.

## 4 Experiments

In this section, we explore the robustness of existing successful OpenIE systems and further analyze the impact of different model architectures on robustness. We first introduce the proposed ROBUST metric, which calculates the robustness performance on a clique, and then extensively evaluate six typical models from three major categories and a large language model ChatGPT. Furthermore, based on the clique structure, we analyze the correlation between the variances of the model performance and the syntactic divergences in cliques.

### 4.1 Evaluation Metrics

The existing widely used CaRB scorer computes pairwise matching scores based on extractions on a sentence. Though accurate, it has rather limitations. We extend this scorer on cliques to calculate the robustness scores.

**The CaRB Metric.** To evaluate the correctness of system tuples, CaRB first creates an all-pair matching table, with each column as a system tuple and each row as a gold tuple, and computes precision and recall scores in each cell. Then, it calculates the overall recall $R$ by averaging the maximum values of all rows and the overall precision $P$ by averaging the one-to-one precisions between system tuples and gold tuples in the order of the best match score to the worst. Finally, the overall $F_1$ is computed with $R$ and $P$.

**The ROBUST Metric.** An OpenIE system is considered robust if it behaves consistently on sentences with the same underlying knowledge meaning but differing syntactic and expressive variations, indicating the preservation of knowledge invariance. Therefore, we naturally calculate the robustness scores of a model on each clique.

Given a clique including $k$ sentences $\mathcal{C} =$

| | | OpenIE4 | ClauseIE | OpenIE5 | RnnOIE | SpanOIE | OpenIE6 |
|---|---|---|---|---|---|---|---|
| | CaRB | **61.0** | 41.66 | 57.39 | 49.26 | 31.31 | 60.86 |
| $P$ | ROBUST | **38.63** | 23.97 | 33.95 | 26.15 | 18.98 | 37.46 |
| | $\Delta$ | ↓22.37 | ↓17.69 | ↓**23.43** | ↓23.11 | ↓12.33 | ↓23.4 |
| | CaRB | 48.25 | 50.16 | 47.48 | 49.46 | 42.21 | **50.54** |
| $R$ | ROBUST | 30.14 | **38.58** | 30.88 | 32.52 | 27.94 | 34.25 |
| | $\Delta$ | ↓18.11 | ↓11.58 | ↓16.60 | ↓**16.94** | ↓14.27 | ↓16.29 |
| | CaRB | 53.88 | 45.51 | 51.97 | 49.36 | 35.95 | **55.22** |
| $F_1$ | ROBUST | 33.86 | 29.57 | 32.34 | 28.99 | 22.60 | **35.78** |
| | $\Delta$ | ↓20.02 | ↓15.94 | ↓19.63 | ↓**20.37** | ↓13.35 | ↓19.44 |

Table 2: The performance of typical OpenIE systems on CaRB and ROBUST benchmarks. The row $\Delta$ represents the the difference between CaRB score and ROBUST score (↓ means the degradation from CaRB). **Bold numbers** refers to the highest score per metric or highest difference per row (i.e. highest $\Delta$ for $P$, $R$ and $F_1$).

$\{s_1, ..., s_k\}$ in ROBUST, we first calculate the $P/R/F_1$ scores of the model on each sentence, and then select the scores from the sentence with the worst $F_1$ as the ultimate robustness scores $P^{robust}/R^{robust}/F_1^{robust}$. As mentioned above, we can compute the pair-wise $P/R/F_1$ scores based on the CaRB scorer.

It is noteworthy that the ROBUST evaluation metric is compatible with existing benchmarks because we calculate on the order of magnitude of one sentence, and we can directly compare our robustness scores with CaRB and others.

### 4.2 Evaluation Models

To exhaustively evaluate the robustness of existing paradigms, we select six typical OpenIE approaches from 3 categories. (1) **Rule-based models**, which adopt linguistic patterns to identify knowledge facts, including *OpenIE4* (Christensen et al., 2011), *ClauseIE* (Del Corro and Gemulla, 2013), and *OpenIE5* (Saha et al., 2017, 2018). (2) **Independent NN-based models**, that train neural networks from scratch with designed architecture, including *RnnOIE* (Stanovsky et al., 2018) and *SpanOIE* (Zhan and Zhao, 2020). (3) **PLM-based models**, that rely on a pre-trained language model usually trained on a large-scale text corpus, including *OpenIE6* (Kolluru et al., 2020a) which introduces a novel iterative grid labeling architecture, which treats OpenIE as a 2-D grid labeling task to produce extractions gradually based on BERT.

We also evaluate the OpenIE performance of ChatGPT. We use the python API interface of gpt-

3.5-turbo version[4] for all experiments. We perform few-shot experiments with manually constructed prompts and sampled demonstrations for CaRB and ROBUST benchmarks. The prompt template is available in Appendix A.3.

### 4.3 Major Results

#### 4.3.1 Results on Typical OIE Models

We run the source code of all baselines on both CaRB and ROBUST and compute the average scores across all samples. All results are shown in Table 2. Note that although the ROBUST scores are calculated in a different environment than CaRB, it still offers a fair comparison due to the calculation manner. Based on the result, we can see that current successive OpenIE systems experience a considerable performance decline on ROBUST across the board. Compared with CaRB, the average degradation for precision, recall, and the $F_1$ score is $20\%$, $15\%$, and $18\%$, respectively. This observation suggests that research on the robustness of existing OpenIE models still needs to be completed, as overly idealized evaluations encourage models to match fixed expressions strictly.

With the concrete comparison of model architectures, we find that the *SpanOIE* model demonstrates a relatively small decrease in all three scores compared to other models, indicating its robustness to syntactic transformations. This result suggests that the extraction strategy of enumerating geometric spans is, to some extent, independent of syntactic drift, making it less susceptible to sentence form transformations.

---

[4]the experimental period is 2023.04.01–2023.05.16.

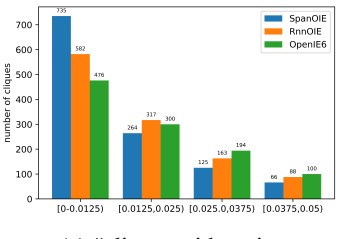

(a) #cliques with variances

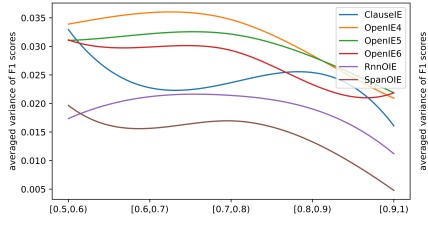

(b) variances with HWS distances

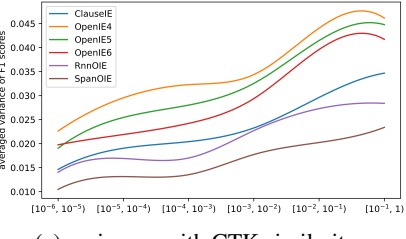

(c) variances with CTK similarites

Figure 5: (a) The distribution of the number of cliques with the variance of $F_1$ scores in each clique. (b) The variance of $F_1$ scores with the values HWS distance. (c) The variance of $F_1$ scores with the values of Convolutional Tree Kernel similarity. The both correlation values are divided into several intervals to avoid abnormal values.

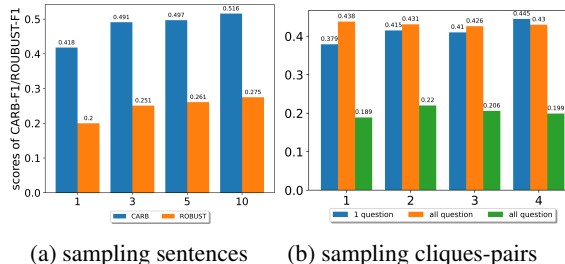

(a) sampling sentences    (b) sampling cliques-pairs

Figure 6: ChatGPT performance on CaRB/ROBUST. (a) The averaged $F_1/F_1^{robust}$ scores on CaRB/ROBUST by randomly sampling demonstrations in CaRB. (b) The averaged $F_1/F_1^{robust}$ scores on ROBUST by randomly sampling 100 clique-pairs and specifying demonstrations and questions from different cliques in each pair.

### 4.3.2 Results on ChatGPT

We evaluate ChatGPT's OpenIE capability on CaRB and ROBUST. We randomly select 1/3/5/10 demonstrations from CaRB, and prompt ChatGPT to extract knowledge tuples by incorporating these demonstrations. We exclude sentences that belong to the same clique as the demonstrations during extraction. The result shows that ChatGPT exhibits impressive capability on CaRB, attaining a 51.6 $F_1$ score in 10-shot setting, comparable to the supervised state-of-the-art model *OpenIE6*. However, it still faces the robustness problem, as evidenced by a decline in the $F_1^{robust}$ score to 27.5 on ROBUST in the same setting.

We also investigate the impact of ChatGPT's performance on the diversity of demonstrations. We first randomly select 100 pairs of cliques $\{(\mathcal{C}_i, \mathcal{C}_j) | (\mathcal{C}_i = (s_i^1, s_i^2, ...)\}^{100}$ from ROBUST. For each sentence in clique $\mathcal{C}_i$, we prompt Chat-GPT by specifying 1/2/3/4 demonstrations from clique $\mathcal{C}_j$. We then calculate the CaRB $F_1$ score for each sentence (*shown in blue*), the average CaRB $F_1$ score for all sentence $(s_i^1, s_i^2, ...)$ (*shown in orange*), and the ROBUST $F^{robust}$ score on all sen-

tence in clique $\mathcal{C}_j$ (*shown in green*). The results in Figure 6b show that the correctness and robustness of ChatGPT can be improved by giving more diversified demonstrations.

### 4.4 Detailed Analysis

In this section, we investigate the coherence among cliques in ROBUST, as well as the variations in model performance across different cliques.

**Is the evaluation of model performance consistent across cliques?** It is necessary to investigate whether our evaluation of the model is consistent across the majority of cliques in order to explore the internal consistency of our data samples. Based on the main results, we calculate the $F_1$ score variance in each clique for three representative models, *Rn-nOIE*, *SpanOIE*, and *OpenIE6*. The distribution of the number of cliques based on variance is depicted in Figure 5a. We find that the majority of cliques exhibit relatively slight variances, indicating a high degree of consistency among robustness cliques. In addition, we sample 11 subsets of interval 100 in ROBUST and calculate the Person's Correlation Coefficient between the average $F^{robust}$ of *Ope-nIE6* on each subset and the number of cliques of each subset. This result is $-0.1480$, indicating a weak correlation between these two factors.

**How does the syntactic divergence affect the performance of models?** Benefiting from the data structure of ROBUST, we can further investigate the effect of syntactic divergence on the performance of models. Concretely, for each clique, we calculate the average HWS/CTK values between all pairs of sentences and the variance of $F_1$ across all sentences. The result is shown in Figure 5[5]. The results indicate a general trend where the variance of model performance decreases with increasing

---

[5]We divide all samples into five intervals and calculate the average variance to avoid abnormal values.

syntactic divergence. Based on the main experiment results, which indicate low performance of models on the overall benchmark, the observed degradation implies a consistent trend of poorer model performance in more open scenarios.

## 5 Related Work

**OpenIE Approaches.** The OpenIE task was first proposed by (Banko et al., 2007) and is a fundamental NLP task. Earlier models focused on statistical or rule-based methods to handle this task (Christensen et al., 2011; Schmitz et al., 2012; Del Corro and Gemulla, 2013; Angeli et al., 2015; Pal et al., 2016; Saha et al., 2017, 2018). Recently, with the rapid development of deep representation learning, many supervised neural models have been proposed for OpenIE. These approaches could be roughly classified into two lines: 1.*Sequence Labeling-based* models. RnnOIE (Stanovsky et al., 2018) applies a BiLSTM transducer, extending deep Semantic Role Labeling models to extract tuples. SenseOIE (Roy et al., 2019) leverages an ensemble of multiple unsupervised OpenIE systems' outputs and the lexical and syntactic information to improve performance. SpanRel (Jiang et al., 2020) represents the OpenIE task in a single format consisting of spans and relations between spans. SpanOIE (Zhan and Zhao, 2020) predicts the candidate relation spans and classifies all possible spans of the sentence as subject or object for each span. Multi$^2$OIE (Ro et al., 2020) first predicts all relational arguments by BERT and then predicts the subject and object arguments associated with each relation using multi-headed attention. OpenIE6 (Kolluru et al., 2020a) provides an iterative grid labeling architecture, which treats OpenIE as a 2-D grid labeling task. 2.*Sequence Generative* models. Neural Open IE (Cui et al., 2018) and Logician (Sun et al., 2018) generate OpenIE extractions by a seq2seq paradigm. IMoJIE (Kolluru et al., 2020b) leverages a BERT-based encoder and generates the next extraction which is fully conditioned on the extractions produced so far.

**OpenIE Benchmarks.** Several benchmark datasets have been proposed to evaluate existing OpenIE approaches. OIE2016 (Stanovsky and Dagan, 2016) developed a method to create a large-scale OpenIE dataset using QA-SRL annotations (He et al., 2015) which was found to be noisy with missing extractions. After that, CaRB (Bhardwaj et al., 2019) and Re-OIE2016 (Zhan and

Zhao, 2020) re-annotated the corpus to improve the dataset's quality for more accurate evaluation. Wire57 (Lechelle et al., 2019) provided high-quality expert annotations, but the size is too small to serve as a comprehensive test dataset with only 57 sentences. DocOIE (Dong et al., 2021) argued that in reality a sentence usually exists as part of a document rather than standalone; the contextual information can help models understand it better and annotate a document-level OpenIE dataset. LSOIE (Solawetz and Larson, 2021) was built by converting the QA-SRL BANK 2.0 dataset (FitzGerald et al., 2018) to OpenIE which had a significant improvement over previous work in terms of data quantity. BenchIE (Gashteovski et al., 2022) created a fact-based benchmark and framework for multi-faceted comprehensive evaluation of OpenIE models in the multi-lingual setting.

Despite the widespread interest in these benchmarks and the related OpenIE approaches provides promising results. However, the traditional peer-to-peer matching-based evaluation can not measure the robustness of those approaches, where the syntax and expression may be various with underlying meaning (Qi et al., 2023). This work significantly fills the gap between traditional metrics and missed robustness evaluation for OpenIE and calls for more efforts in this research area.

## 6 Conclusion and Future Work

In this work, we propose ROBUST, a large-scale human-annotated OpenIE benchmark consisting of 1272 robustness testing cliques, where each clique contains sentences with different syntactic and expressive variations while conveying the same underlying knowledge meaning. We introduce our methodology for constructing the benchmark, including a syntactically and expressively diverse paraphrase generation, and a two-stage manual annotation. A comprehensive analysis is then performed to demonstrate the consistency of the proposed data with the real world. We finally perform extensive experiments on existing successive models as well as a representative large language model, and the results show that the robustness of existing methods is far from satisfied. The further detailed analysis demonstrates the substantial internal coherence of our benchmark, providing inspiration for the future development of robustness benchmarks.

## Acknowledgement

We sincerely appreciate the three reviewers from ACL2023, who provided thorough reviews and suggestions during the initial submission of this work. This work is supported by the National Natural Science Foundation of China (No. 62277033). It also got partial support from National Engineering Laboratory for Cyberlearning and Intelligent Technology, and Beijing Key Lab of Networked Multimedia. This work is also supported by a grant from the Institute for Guo Qiang, Tsinghua University (2019GQB0003) and the NSFC Youth Project (62006136).

## 7 Limitations

We have presented a dataset with metrics to evaluate the robustness of OpenIE models in this paper. However, there are still several limitations that need to be improved in further study. First, there are a few studies exploring the pre-trained language models to perform zero-shot information extraction with advantages. To the lack of open source code, we have not explored the robustness performance of these zero-shot models. Second, we think the robustness problem generally exists in the NLP community, we remain the extensive study of robustness examination for more domains and models in future works.

## 8 Ethic Consideration

There are two major considerations for conducting the evaluation of our proposed new benchmark. First, the source sentences are selected as same as CaRB, the original dev and test splits of OIE2016 in the open domain source of Wall Street Journal text and Wikipedia. All these data files are leveraged for the research purpose, and the result will be publicly available. Second, the annotators in this research are paid a salary higher than the market average and further allowed to choose flexible working time for human rights. For data utilization, we will make all annotation results publicly available under the CC BY-SA 4.0 license (free for research use).

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

# A Appendix

## A.1 Annotation Details

We have the following detailed annotation information. **Who**: For Task1 and Task2, we employed two separate annotation teams consisting of 6 and 9 students respectively, who are all majoring in CS at universities. We ensured their professionalism through the tutorials and a final examination. **Where**: As both tasks are easy to read and write for annotators, we distributed the data directly without using a special annotation platform. **Quality**: We adopted a batched iterative annotation and evaluation process to ensure that the sampling accuracy is above 90%. **License**: We will release all annotation results under the CC BY-SA 4.0 license (free for research use).

### A.1.1 Paraphrase Annotation Process

The goal of paraphrase annotation is to correct the automatically generated sentences from the models based on human intelligence. Overall, we adopt an iterative step of combining human annotation paired with expert evaluation to ensure accuracy and efficiency. In each iteration, at least three human workers who are fluent in English reading and writing annotate a batch of samples, and then two domain experts will check the annotation results on a random sample of 40% of the batch. The batch annotations will be accepted until the validation accuracy is greater than 90%. For the annotation of each paraphrase, the annotators are asked to correct the sentence with syntactic, phrasal, or semantic-different mistakes against the original sentence.

### A.1.2 N-tuples Annotation Process

We leverage the same iterative annotation strategy with the paraphrase annotation for OpenIE N-tuples annotation. In particular, we design an annotation flowchart for the workers according to the similar process in CaRB, by dividing the task into 4 steps: (1) identifying the relation, (2) identifying the arguments for that relation, and (3) optionally identifying the location and time attributes for the tuple. The same validation meaner with the paraphrase annotation is adopted to reach each acceptable annotation batch.

---

**Algorithm 1 HWS Distance**

**Input:** Constituency parses $T_1, T_2$ of sentences $s_1, s_2$, pruning height $h$, discount factor $\alpha$

**Output:** Syntactic distance $d$ between $s_1, s_2$

1: Get trees $T_1^h, T_2^h$ pruned at height $h$, and their level-order traversal sequences $q_1, q_2$
2: Initialize total length and count $l = 0; m = 0$
3: $A[i][1] = 1$ if $q_1[i] == q_2[1], i = 1, ..., q_{1.len}$
4: $A[1][j] = 1$ if $q_1[1] == q_2[j], j = 1, ..., q_{2.len}$
5: $J[1] = 1$ if $q_1[i] == q_2[1], i = 1, ..., q_{1.len}$
6: $I[1] = 1$ if $q_1[1] == q_2[j], j = 1, ..., q_{2.len}$
7: **for** $i = 2 \to q_{1.len}$ **do**
8:      **for** $j = 2 \to q_{2.len}$ **do**
9:          **if** $q_1[i] == q_2[j]$ **then**
10:              **if** $A[i-1][j-1] \geq I[i]$ && $A[i-1][j-1] \geq J[j]$ **then**
11:                  $A[i][j] = A[i-1][j-1] + 1$
12:                  $I[i] = A[i-1][j-1] + 1$
13:                  $J[i] = A[i-1][j-1] + 1$
14:              **end if**
15:          **else**
16:              $A[i][j] = 0$
17:              **if** $A[i-1][j-1] > 1$ **then**
18:                  $l = l + A[i-1][j-1] \times \alpha^m$
19:                  $m++$
20:              **end if**
21:          **end if**
22:      **end for**
23: **end for**
24: **if** $A[i][j] \geq 1$ **then**
25:      $l = l + A[i][j] \times \alpha^m$
26: **end if**
27: Return $1 - l/min(q_{1.len}, q_{2.len})$

---

## A.2 Algorithms Details

### A.2.1 Hierarchically Weighted Syntactic Distance Algorithm (Revised)

The revised Hierarchically Weighted Syntactic Distance Algorithm (HWS distance) is shown in algorithm 1. We fix the over-counting problem for repeated consecutive spans while preserving the efficiency with the same time complexity in the original work (Qi et al., 2023).

### A.2.2 Diversified Filtering Process

We perform diversified filtering based on BLEU scores between all pairs of sentences in each set of generated paraphrases to maintain the most diverse paraphrases. For example, given the generated paraphrases following:

| $ori$ | In 1840, he was appointed to command his regiment, a post he held for nearly fourteen years. |
|---|---|
| $p_1$ | 1840, the regiment's commander, which he held for nearly 14 years. |
| $p_2$ | In 1840 he took command of the regiment and held it for nearly 14 years. |
| $p_3$ | When he was 14 years old , he became a member of the regiment . |
| $p_4$ | 1840, the command of the regiment, which he held for nearly 14 years. |
| $p_5$ | The regiment, then, in 1840, the rank of captain, which he held for nearly 14 years. |

Table 3: The original sentence $ori$ with 5 paraphrases $p_1 \sim p_5$.

As shown in Figure 7, we first calculate the BLEU scores between all pairs of paraphrases (shown on the edges). We then find the two sentences $p_1, p_4$ with the maximum BLEU score. Because the lengths of these two sentences are larger than $2/3$ of the original sentence, we then calculate the summation of scores from each of them to all other sentences which results $sum(p_1, p_{/1}) = 136.9$ and $sum(p_1, p_{/1}) = 158.7$, and remove the sentence $p_4$ that has larger summation score. We repeat the strategy above to remove the sentence $p_1$ and obtain 3 expressively diverse paraphrases.

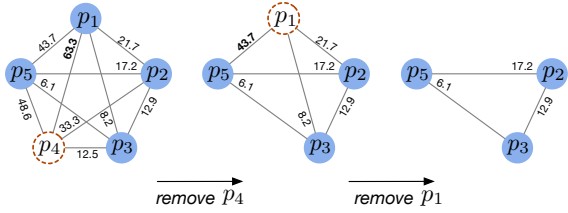

Figure 7: By performing the diversified filtering, 3 paraphrases $p_2, p_3, p_5$ is maintained.

### A.3 Prompts and Analysis for ChatGPT

#### A.3.1 Prompt Design

We create a prompt template for the task of OpenIE to query the ChatGPT. An example of a 1-shot prompt is shown in Figure 8, where the highlighted demonstration and the variable <sentence> can be replaced with specified examples.

#### A.3.2 Performance with Syntactic Correlations

In this section, we further investigate the correlation between the model performance and syntactic distance of demonstrations and questions for the ChatGPT model. We first randomly sample a set of 100 pairs of cliques $\{(\mathcal{C}_1^i, \mathcal{C}_2^i) | i = 1, ..., 100\}$

*prompt* = "Open information extraction requires the extraction of all relations in the sentence, i.e., predicates, the subjects and objects corresponding to these relations, and the possible time and place thesis elements. For example, in the sentence: *Watson, who was elected by his caucus in November 1998, has served as Minority Leader since then.*

From this sentence, the following tuple can be extracted: *(was elected by, Wastson, his caucus in November 1998); (has served as, Wastson, Minority Leader, since then)*

In these tuples, we always put the predicate first, the second is the subject corresponding to the predicate, the third is the object corresponding to the predicate (if there is none, it is not labeled), and the last two are time and place in that order, which can be omitted if there is none.

Please follow the example above and extract all the relational tuples in the following sentence: *<sentence>*

Please show the results in one line strictly in the form of the results above"

Figure 8: The 1-shot prompt to ChatGPT for the OpenIE task, where the <sentence> corresponds to the query sentence.

in ROBUST. Then for each pair, we select all examples in clique $\mathcal{C}_1^i$ as demonstrations and select all sentences in $\mathcal{C}_2^i$ as questions to calculate the $F_1^{robust}$-score. For syntactic correlations, we first calculate the averaged value $a_i$ between question $i$ and all sentences in $\mathcal{C}_1$ and further calculate the average on $(a_1, a_2, ...)$ as the final correlation on current clique-pairs. We divide the scores into several intervals and compute the average value in each corresponding interval to avoid abnormal values. The results based on both implementations of HWS distance and Tree Kernel similarity as the syntactic correlation are shown in Figure 9.

In the left figure of the result, we can see that the $F_1^{robust}$-score of the model gradually increases as the average syntactic similarity of the two cliques increases. The same observation is also shown in the right figure with the averaged syntactic distance between two cliques. These results suggest that ChatGPT is sensitive to the syntactic distribution between questions and demonstrations and that giving demonstrations with similar syntactic distribution enhances the effectiveness of ChatGPT.

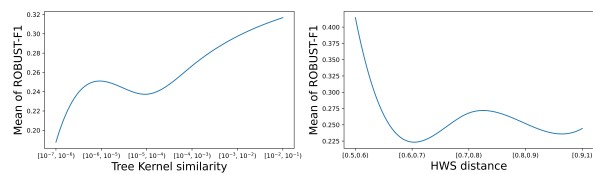

Figure 9: The $F_1^{robust}$ scores of OpenIE6 model with syntactic correlations between clique-pairs.

### A.4 Error Analysis for OIE Systems

We conduct error analysis for three typical OpenIE models OpenIE4, SpanOIE, and OpenIE6 on a robustness clique. The model predictions with the CaRB and ROBUST scores are shown in Table 4.

First, we can see that the sentences in the clique exhibit a significant syntactic and expressive divergence. It implies that the constructed data source satisfies the expectation. Second, we find all sentences in the clique have more than one extraction, while the OpenIE4 and OpenIE6 models predict the extractions insufficiently, which causes a lower recall. On the other hand, the SpanOIE model outputs predictions by enumerating all possible geometric spans, which build sufficient outputs regardless of syntactic features. This architecture offers SpanOIE a consistent performance.

| | Sentences & Extractions | CaRB | ROBUST |
|---|---|---|---|
| *Clique*$_1$ | ♦ They can be relieved only by changing that system, not by pouring Western money into it.
—->(can be relieved, They)
—-> (by changing, They, that system) | / | / |
| | ♣ Only a change in the system, not an injection of Western money can relieve them.
—-> (can relieve, Only a change in the system, them)
—-> (An injection of Western money, cannot relieve, them)
—-> (can be changed, the system) | / | / |
| | ■ Instead of pouring money from the West, changing the system is the only way to relieve them.
—-> (is, Changing the system, the only way to relieve them)
—-> (is not, Pouring money from the West, the only way to relieve them)
—-> (can relieve, Only a change in the system, them) | / | / |
| | ♠ What can relieve them is only to change that system, not to put money from the West into it.
—-> (can relieve, only to change that system, them)
—-> (cannot relieve, put money from the West into it, them) | / | / |
| OpenIE4 | ♦ (can be relieved, They) | 0.486 | 0.119 |
| | ♣ (can relieve, Only a change in the system, not, them) | 0.378 | |
| | ■ (is, changing the system, the only way to relieve them) | 0.553 | |
| | ♠ (is, What can relieve them, only to change that system, not to put money from the West into it) | 0.119 | |
| SpanOIE | ♦ (can be relieved only by, They, changing that system),
(can be relieved not by pouring), They, Western money into that system),
(is of, money, the West) | 0.597 | 0.157 |
| | ♣ (can relieve, Only a change in the system, them.),
(cannot relieve, An injection of Western money, them.),
(can be changed, the system | 0.414 | |
| | ■ (is, Changing the system, the only way to relieve them.),
(is not, Pouring money from the West, the only way to relieve them.) | 0.359 | |
| | ♠ (can relieve, only to change that system, them),
(cannot relieve, put money from the West into it, them) | 0.157 | |
| OpenIE6 | ♦ (They, can be relieved, only by changing that system )
(They, only by changing, that system) | 0.48 | 0.334 |
| | ♣ (Only a change in the system , not, can relieve, them) | 0.421 | |
| | ■ (changing the system, is, the only way to relieve them) | 0.533 | |
| | ♠ (What can relieve them, only to change that system , not to put money from the West into it) | 0.334 | |

Table 4: An error analysis for model predictions with the $F_1/F_1^{robust}$ scores of two benchmarks.