# OpenReview forum: "Preserving Knowledge Invariance: Rethinking Robustness Evaluation of Open Information Extraction"
_EMNLP/2023/Conference — EMNLP 2023 Main_

### Official Review · Reviewer_2rx1 · 2023-07-24

**Soundness:** 4

**Excitement:**

4: Strong: This paper deepens the understanding of some phenomenon or lowers the barriers to an existing research direction.

**Missing References:**

Studies related to robustness, in the sense of consistency across paraphrases, were not mentioned as related work. Some seminal papers:

Di Jin, Zhijing Jin, Joey Tianyi Zhou, Peter Szolovits:
Is BERT Really Robust? A Strong Baseline for Natural Language Attack on Text Classification and Entailment. AAAI 2020: 8018-8025

Yanai Elazar, Nora Kassner, Shauli Ravfogel, Abhilasha Ravichander, Eduard H. Hovy, Hinrich Schütze, Yoav Goldberg:
Measuring and Improving Consistency in Pretrained Language Models. Trans. Assoc. Comput. Linguistics 9: 1012-1031 (2021)

**Paper Topic And Main Contributions:**

This paper introduces a new benchmark (a dataset and a metric) named ROBUST for evaluating robustness in OpenIE systems. It attempts to answer the question: will the performance of existing OpenIE systems drop when provided paraphrases of test sentences that express the same relation differently?

A systematic method to generate paraphrases that are diverse in both syntax and expressions was used to derive the new dataset from CaRB. Human annotations were involved to ensure the quality of the dataset.

Evaluation was performed on a wide range of existing methods.

**Questions For The Authors:**

Question A: In Figure 5 and Section 4.4, is there a relation between variance in F1 within cliques, and the general robustness of OpenIE systems? If so, does the detailed analysis imply that the cliques with more diverse, out-of-distribution paraphrases will actually lead to higher consistency (and hence higher robustness)?

Question B: Are the new paraphrases in ROBUST dataset as a whole different from CaRB, in terms of syntax and expressions? Section 3.2 only compares this within ROBUST.

Question C: Do you have insights on the root cause for the accuracy degradation? Was that because of overfitting, or the intrinsic hardness of certain paraphrases?

Question D: Line 190: What’s the motivation behind the removal?

**Reasons To Accept:**

The paper is well-written and easy to follow. The process for creating the dataset is systematic and rigorous.

The paper confirms an important, unsolved, and ubiquitous issue in existing OpenIE systems, and also provides a reasonable benchmark to evaluate the issue. These could bring about future research ideas and improved OpenIE systems.

If the ROBUST dataset is released, the community will have a high-quality resource for evaluating robustness in OpenIE. The automated method and manual annotation process for paraphrasing also have the potential to benefit research related to paraphrasing and language education.

**Reasons To Reject:**

The proposed metric (lowest F1 score among paraphrases) does not seem particularly appropriate for evaluating “robustness”. The authors defined robustness as “behaves consistently on (paraphrases)”, but the min can not reflect variance / inconsistency. A counterexample: a really bad OpenIE system that **consistently** get a very low F1 score on any sentence will have a very low ROBUST score, but it’s performance is actually consistent.

Some findings in “detailed analysis” are conflicting with the main conclusion and no explanations were provided. Figure 5 suggests that the more diverse the paraphrases are, the more consistency there is. In contrast, the main argument of the paper is that different form leads to degraded accuracy, which means more variance because some F1 scores are lower.

**Reproducibility:**

5: Could easily reproduce the results.

**Reviewer Confidence:**

4: Quite sure. I tried to check the important points carefully. It's unlikely, though conceivable, that I missed something that should affect my ratings.

**Typos Grammar Style And Presentation Improvements:**

Line 297, 517 and multiple other occurrences: from ACLPUB paper formatting guidelines:
> Refrain from using full citations as sentence constituents. Instead of “(Gusfield, 1997) showed that … In (Gusfield, 1997), …” write “Gusfield (1997) showed that … In Gusfield (1997), …”

Line 323 and Figure 3: I suppose “x-axis” in the caption means the horizontal axis? It is a bit ambiguous. Perhaps having axis labels is better.

Line 417: python -> Python

Line 418 and footnote 4: OpenAI provides official versioning for the models behind their API, for example “gpt-3.5-turbo-0613”

---

> ### Author Rebuttal · Authors · 2023-08-29
>
> We appreciate the reviewer for the careful review and helpful suggestions. In response to the questions you raised, we have made the following explanations and revisions:
>
>
> > The proposed metric (lowest F1 score among paraphrases) does not seem particularly appropriate for evaluating “robustness”. The authors defined robustness as “behaves consistently on (paraphrases)”, but the min can not reflect variance / inconsistency. A counterexample: a really bad OpenIE system that consistently get a very low F1 score on any sentence will have a very low ROBUST score, but it’s performance is actually consistent.
>
> A1:  The worst performance indicates the lower bound of the model when applying to the realistic scenarios, which is more reliable for estimating the robustness of models in practical applications and is inline with some existing literatures [1]. We highly acknowledge the counterexample you have provided. In fact, as mentioned in line 20, we believe that a model needs to perform accurately and consistently on a clique in order to be considered robust. Thank you for your suggestion. We found that there was ambiguity in describing the robustness metric in line 379, and we will revise the description to “behave consistently accurate ..”. to make the statement clear.
>
> > Some findings in “detailed analysis” are conflicting with the main conclusion and no explanations were provided. Figure 5 suggests that the more diverse the paraphrases are, the more consistency there is. In contrast, the main argument of the paper is that different form leads to degraded accuracy, which means more variance because some F1 scores are lower.
>
> A2: We also observed and analyzed the phenomenon shown in Figure 5(a/b). By examine the cliques with high syntactic variance and the models’ performance on these cliques, we found that (1) the paraphrases tend to deviate more from the original sentence in syntactic structures and expressive forms, and (2) the models show consistent low performance (i.e., low variance) across all paraphrases (an average of 3 paraphrases in each clique). Therefore, we hypothesize that this accounts for the slight trend reflected in the Figure 5(a/b).
>
> > Question A: In Figure 5 and Section 4.4, is there a relation between variance in F1 within cliques, and the general robustness of OpenIE systems? If so, does the detailed analysis imply that the cliques with more diverse, out-of-distribution paraphrases will actually lead to higher consistency (and hence higher robustness)?
>
> Answer to question A: Although we have generated as comprehensive syntactic and expressive variations as possible during preparation and annotation, the diversity within cliques still has limitations compared to the real-world. Therefore, the variances in model performance within cliques cannot rigorously reflect the robustness of models, but rather reflects the inter-consistencies of the cliques. This is in hopes of promoting the development of an effective data structure for other robustness evaluations. In fact, this is also one of the reasons that we consider the lowest performance as the final robustness score in the definition.
>
> > Question B: Are the new paraphrases in ROBUST dataset as a whole different from CaRB, in terms of syntax and expressions? Section 3.2 only compares this within ROBUST.
>
> Answer to question B: Based on the syntactically controlled generation and diversified filtering, we devoted ourselves to ensure the differences of generated sentences from CaRB. We manually checked the generation results based on randomly sampling on both two stages, and we found that most of the paraphrases exhibit the obvious syntactic differences (e.g, transformation from active to passive voice or the inclusion of clauses) in stage 1, and the same paraphrases as the original sentence are further rewritten into different forms in syntax and expressions at stage 2.
>
> > Question C: Do you have insights on the root cause for the accuracy degradation? Was that because of overfitting, or the intrinsic hardness of certain paraphrases?
>
> Answer to question C: The conclusion from Figure 1 in RobustOIE [2] shows that the OpenIE model performance exhibits a significant degradation as the syntactic similarity between the training set OIE2016 and testing set CaRB decreases. We hypothesis that the existing models trained on the OIE2016 which originated from Wiki and WSJ newswires with fixed semantic-syntactic pairs may suffer from the generalization issue to syntactic/expressive variants, and the existing testset with the design is difficult to validate this generalization performance. Based on our analysis, we suggest that the more diverse training corpora incorporating domain-specific priors (e.g, syntactic and semantic knowledge) may be a good solution to mitigate the domain bias for the adaption of models.
>
> > Question D: Line 190: What’s the motivation behind the removal?
>
> Answer to question D: Based on the observation on the generated paraphrases, we find that most of the low-quality cases tend to be a paraphrase of partial content of the original sentence. By statistics on randomly sampled generations, we remove the generated paraphrases that the lengths are less than 2/3 of the original sentence.
>
>
> Reference:
> [1] Zhong, Victor, et al. "Romqa: A benchmark for robust, multi-evidence, multi-answer question answering." arXiv preprint arXiv:2210.14353 (2022).
> [2] Qi, Ji, et al. "Syntactically Robust Training on Partially-Observed Data for Open Information Extraction." In EMNLP 2022.

---

### Official Review · Reviewer_snS1 · 2023-08-02

**Soundness:** 4

**Excitement:**

4: Strong: This paper deepens the understanding of some phenomenon or lowers the barriers to an existing research direction.

**Paper Topic And Main Contributions:**

In this work, the authors introduce ROBUST, a benchmark for the OpenIE task, along with a specialized metric to evaluate the robustness (to paraphrasing) of the OpenIE models. The authors argue that conventional evaluation benchmarks for the OpenIE task do not measure the robustness of the models in a real-world scenario. As an example, the authors provide a sentence s from the CaRB benchmark and show that if this sentence is paraphrased, i.e., the same semantic meaning but a different distribution of tokens, then the state-of-the-art models such as OpenIE6 fail to identify correctly the knowledge tuples from the paraphrased sentence. Therefore, instead of evaluating the pair-wise exact match between the model-extracted knowledge tuple and the ground-truth knowledge tuple, the ROBUST benchmark proposes to assess at a clique level; Given a query sentence q, a clique of semantically equivalent yet paraphrased sentences is constructed via a carefully designed paraphraser model (with diversified syntactic sampling and filtering strategies). Evaluating at a clique level, the authors claim that one can assess the performance variations (and, in turn, model robustness) due to distributional drifts within the paraphrased sentences. This analysis provides a better understanding of the model and is also compatible with existing benchmarks for the OpenIE task.

For generating the paraphrases, the authors start with the existing CarB benchmark and use the AESOP paraphrasing model (trained on a parallel annotated dataset) and the ParaNMT-50M corpus to identify ten syntactically varying paraphrases for each sentence within the CarB benchmark. A heuristic search strategy is applied to ensure that the syntactic structures and the expressive forms of the paraphrased sentences are diverse from the original sentence. The resulting paraphrases are then subject to a two-stage annotation pipeline done manually for sentence correction and structured knowledge extraction. For the sentence correction step, the annotators look at correcting grammar, replacing phrases (if required), or rewriting the entire sentence if it's semantically different from the original. For the Knowledge annotation step, human annotators are asked to annotate N-ary knowledge tuples on the paraphrases generated previously.

Subsequently, the authors provide a comprehensive analysis of the ROBUST benchmark. To be exact, they quantify the syntactic similarity/distance of the sentences in a given clique via the HWS distance/Convolutional Tree Kernel similarity functions and demonstrate that the sentences in a clique are syntactically divergent. For experimental evaluations, the authors introduce the ROBUST metric.
The ROBUST metric is calculated on each clique of sentences instead of a single sentence. It uses the CarB scorer to evaluate the model output, i.e., P/R/F1 scores on each sentence. It selects the scores from the sentence with the worst F-measure as the ultimate robustness P/R/F score. This makes the ROBUST metric comparable to the CaRB.

For empirical evaluations, the authors evaluate six typical OpenIE models from three categories: Rule-based, Independent neural network-based, and Pretrained language model-based. They observe that all six models experience a considerable decline in the ROBUST benchmark, thereby enhancing the need to build OpenIE systems that are robust to paraphrasing.

Additionally, the authors evaluate the OpenIE capability of ChatGPT by providing 1/3/5/10 examples from the CarB benchmark as prompts. While ChatGPT under the ten-shot setting performs comparably to the supervised state-of-the-art OpenIE6 model, it too registers a decline in the performance on the ROBUST benchmark. Moreover, the authors also provide detailed analyses investigating the clique coherence in the ROBUST benchmark and the variations in model performance across different cliques.

**Reasons To Accept:**

1. This work portrays the robustness issue in evaluating OpenIE models, which could be significant when using such models for inferencing in a zero/few-shot setting using real-world data. This problem has been well introduced and adequately motivated.

2. The subsequent sections detailing the creation of the ROBUST benchmark and the shortcomings of current state-of-the-art models regarding robustness are explained well and are easy to follow. The analysis sections provide detailed insights into the benchmark properties; the variation of a model's performance across different cliques could be used for a detailed error analysis regarding the model's generalization capabilities.

3. Release of the ROBUST benchmark, which can serve as a test-bed suite for the robustness evaluation of the OpenIE models.

**Reasons To Reject:**

None

**Reproducibility:**

4: Could mostly reproduce the results, but there may be some variation because of sample variance or minor variations in their interpretation of the protocol or method.

**Reviewer Confidence:**

4: Quite sure. I tried to check the important points carefully. It's unlikely, though conceivable, that I missed something that should affect my ratings.

---

> ### Author Rebuttal · Authors · 2023-08-29
>
> Thank you for taking the time to review our paper. We are grateful for your recognition and encouragement, which motivates us to further explore and address the challenges facing the OpenIE and and NLP community, and make more meaningful contributions in the future.

---

### Official Review · Reviewer_59er · 2023-08-03

**Soundness:** 4

**Excitement:**

4: Strong: This paper deepens the understanding of some phenomenon or lowers the barriers to an existing research direction.

**Paper Topic And Main Contributions:**

To evaluate the robustness of open information extraction (OIE) models on sentences with different syntactic structures, this paper introduces a new dataset.
The dataset is based on cliques, where sentences with the same semantic meaning but in different syntactic structures are in the same clique.
Syntactic-guided paraphrasing is applied to build the dataset with manual annotation as post-correction.
The experimental results demonstrate that current popular models suffer from syntactic shifts, and this dataset is able to be a benchmark to evaluate the robustness.

**Questions For The Authors:**

- Question A: The syntactic-aware paraphrasing process is very valuable as a common data augmentation method for other tasks. When performing error corrections in the paraphrase annotation, do you have statistics on the percentage of each type of error?
- Question B: In line 276-277, I agree that wiki- or newswire-based texts may bring distribution bias. But I do not think the syntactic paraphrasing could mitigate the domain bias in a more realistic scenario. After all, the contents' semantics are not changed, they are just be paraphrased to another form.
- Question C: As illustrated in Figure 1, the OIE model may generalize pooly on sentences in different types of syntactic structures. To substantiate the problem, have you analyzed a specific dataset (e.g. CaRB) to see if there are obvious syntactic patterns or characteristics?
- Question D: What if we use a pre-trained paraphrasing model (or ChatGPT) to directly generate paraphrased pairs without explicit syntactic information guidance? Would the syntactic distance/similarity be larger or smaller? Is the syntactic-guided paraphrasing better than vanilla free-form paraphrasing?
- Question E: Comparing with the performance drop (~20% F1 scores) on ROBUST, the variance (<5%) seems to be trivial. So why do you choose the worst F1 score as the ROBUST metric instead of the averaged or the median scores? Could you provide more explanations on the metric score design?

**Reasons To Accept:**

- The syntax-guided paraphrasing process is a valuable data augmentation method that could be utilized in many tasks.
- The new dataset could provide a quantitive robustness analysis for current OIE models, and the metric is compatible with other benchmarks.

**Reasons To Reject:**

- The syntax-guided paraphrasing is intuitive, but it could be better if free-form paraphrasing is applied for comparison.
- The motivation is clear, but the evidence to the syntax-disturbance problem may be not sufficient.

**Reproducibility:**

3: Could reproduce the results with some difficulty. The settings of parameters are underspecified or subjectively determined; the training/evaluation data are not widely available.

**Reviewer Confidence:**

4: Quite sure. I tried to check the important points carefully. It's unlikely, though conceivable, that I missed something that should affect my ratings.

**Typos Grammar Style And Presentation Improvements:**

- It would be better if the numbers in Table 1 could be right-aligned.

---

> ### Author Rebuttal · Authors · 2023-08-29
>
> Thank you for the effort you have invested in reviewing our work and the helpful suggestions. We are honored to address the questions you have raised:
>
>
> > The syntax-guided paraphrasing is intuitive, but it could be better if free-form paraphrasing is applied for comparison.
>
> A1: During the data preparation, we have indeed experimented with both two types of models: (1) the free-form paraphrasing models, including SOW-REAP [1] and UPSA [2], and (2) the syntactically-controllable paraphrasing models, including SGCP [3] and AESOP. By applying the models to our data, we find that the former ones tend to generate paraphrases with minor changes compared to the original sentences (low syntactic distance and high BLEU score), usually exhibiting the replaces of words or phrases of different surface forms that do not satisfy our realistic assumption. So we choose the best performant model AESOP based on our experience.
>
> > The motivation is clear, but the evidence to the syntax-disturbance problem may be not sufficient.
>
> A2: The Figure 1 from RobustOIE [4] demonstrated that the OIE model performance exhibits a significant degradation as the syntactic similarity between the training set and testing sentences decreases, supporting the syntax-disturbance problem. We mentioned this conclusion in line 58  without explaining it in details, and we will revise the statement to clarify the disturbance phenomenon.
>
> > Question A: The syntactic-aware paraphrasing process is very valuable as a common data augmentation method for other tasks. When performing error corrections in the paraphrase annotation, do you have statistics on the percentage of each type of error?
>
> Answer to question A: We have defined the three types of mistakes in the guideline to enable annotators to efficiently and logically correct the sentence. However, as our goal is robustness benchmark of OIE, we have not labeled these error types. We appreciate your valuable suggestion and will perform sampling on the generated sentences to observe the error cases of the paraphrase generation model in future.
>
> > Question B: In line 276-277, I agree that wiki- or newswire-based texts may bring distribution bias. But I do not think the syntactic paraphrasing could mitigate the domain bias in a more realistic scenario. After all, the contents' semantics are not changed, they are just be paraphrased to another form.
>
> Answer to question B: We agree with your perspective that the content's semantics of the dataset remain unchanged after paraphrasing. As the main experiment from RobustOIE [5] shows that syntactically diverse data-augmentation will partially improve the OIE performance on the specific test-set,  we suggest that the more diverse training corpora incorporating domain-specific priors (e.g, syntactic and semantic knowledge) may be a good solution to mitigate the domain bias for the adaption of models.
>
> > Question C: As illustrated in Figure 1, the OIE model may generalize poorly on sentences in different types of syntactic structures. To substantiate the problem, have you analyzed a specific dataset (e.g. CaRB) to see if there are obvious syntactic patterns or characteristics?
>
> Answer to question C: To analyze the syntactic patterns for a specific dataset (e.g., CaRB), one of the most efficient ways is to perform the K-means clustering based on the proposed syntactic metric of HWS-distance. We did this on CaRB and found that the cluster with the largest number of sentences has the smallest average syntactic distance to the OIE2016 training set, and the samples in the cluster tend to have more formal and declarative sentence structures (e.g., one of the cluster centers is (ROOT(S(NP(NNP)(NNP))(VP(VBZ)(ADJP))(. )))).
>
> > Question D: What if we use a pre-trained paraphrasing model (or ChatGPT) to directly generate paraphrased pairs without explicit syntactic information guidance? Would the syntactic distance/similarity be larger or smaller? Is the syntactic-guided paraphrasing better than vanilla free-form paraphrasing?
>
> Answer to question D: Kindly refer to the explanation in A1.
>
> > Question E: Comparing with the performance drop (~20% F1 scores) on ROBUST, the variance (<5%) seems to be trivial. So why do you choose the worst F1 score as the ROBUST metric instead of the averaged or the median scores? Could you provide more explanations on the metric score design?
>
> Answer to question E: The worst performance indicates the lower bound of the model when applying to realistic scenarios, which is more reliable for estimating the robustness of the model in practical applications and is inline with some existing literatures [5]. However, we acknowledge and appreciate your concern, so our scoring code is developed to support the reporting of variances of model performance for more flexible requirements.
>
>
> Reference:
> [1] Goyal, Tanya, and Greg Durrett. "Neural Syntactic Preordering for Controlled Paraphrase Generation." In ACL 2020.
> [2] Liu, Xianggen, et al. "Unsupervised Paraphrasing by Simulated Annealing." In ACL 2020.
> [3] Kumar, Ashutosh, et al. "Syntax-guided controlled generation of paraphrases." In TACL 2020.
> [4] Qi, Ji, et al. "Syntactically Robust Training on Partially-Observed Data for Open Information Extraction." In EMNLP 2022.
> [5] Zhong, Victor, et al. "Romqa: A benchmark for robust, multi-evidence, multi-answer question answering." arXiv preprint arXiv:2210.14353 (2022).

---

### Meta-Review · Area_Chair_t6dr · 2023-09-15

**Recommendation:** 5

**Metareview:**

This paper introduces a new benchmark for evaluating the robustness of OIE systems to distributional drifts surfaced as syntactic structure variations of semantically similar statements.

The reviewers unanimously acknowledge the importance of the proposed resource, the quality of the dataset construction and the in-depth analysis of the experiments. Reviewers 59er and 2rx1 raised some issues and posed some interesting questions that were successfully addressed by the authors and the clarifications will further enhance the manuscript.

---

### Decision · Program_Chairs · 2023-10-07

**Decision:**

Accept-Main

**Comment:**

This paper introduces a new benchmark for evaluating the robustness of OIE systems to distributional drifts surfaced as syntactic structure variations of semantically similar statements.

The reviewers unanimously acknowledge the importance of the proposed resource, the quality of the dataset construction and the in-depth analysis of the experiments. Reviewers 59er and 2rx1 raised some issues and posed some interesting questions that were successfully addressed by the authors and the clarifications will further enhance the manuscript.